# Millimeter-Wave Radar and Vision Fusion Target Detection Algorithm Based on an Extended Network

Chunyang Qi [1], Chuanxue Song [2], Naifu Zhang [3], Shixin Song [3], Xinyu Wang [2] and Feng Xiao [1],*

1    State Key Laboratory of Automotive Simulation and Control, Jilin University, Changchun 130022, China
2    College of Automotive Engineering, Jilin University, Changchun 130022, China
3    School of Mechanical and Aerospace Engineering, Jilin University, Changchun 130022, China
*    Correspondence: xiaofengjl@jlu.edu.cn; Tel.: +86-178-0808-1932

**Abstract:** The need for a vehicle to perceive information about the external environmental as an independent intelligent individual has grown with the progress of intelligent driving from primary driver assistance to high-level autonomous driving. The ability of a common independent sensing unit to sense the external environment is limited by the sensor's own characteristics and algorithm level. Hence, a common independent sensing unit fails to obtain comprehensive sensing information independently under conditions such as rain, fog, and night. Accordingly, an extended network-based fusion target detection algorithm for millimeter-wave radar and vision fusion is proposed in this work by combining the complementary perceptual performance of in-vehicle sensing elements, cost effectiveness, and maturity of independent detection technologies. Feature-level fusion is first used in this work according to the analysis of technical routes of the millimeter-wave radar and vision fusion. Training and test evaluation of the algorithm are carried out on the nuScenes dataset and test data from a homemade data acquisition platform. An extended investigation on the RetinaNet one-stage target detection algorithm based on the VGG-16+FPN backbone detection network is then conducted in this work to introduce millimeter-wave radar images as auxiliary information for visual image target detection. We use two-channel radar and three-channel visual images as inputs of the fusion network. We also propose an extended VGG-16 network applicable to millimeter-wave radar and visual fusion and an extended feature pyramid network. Test results showed that the mAP of the proposed network improves by 2.9% and the small target accuracy is enhanced by 18.73% compared with those of the reference network for pure visual image target detection. This finding verified the detection capability and algorithmic feasibility of the proposed extended fusion target detection network for visually insensitive targets.

**Keywords:** intelligent driving vehicle; multi-sensor fusion; object detection; extended network





## 1. Introduction

The demand for autonomous vehicle sensing capability has gradually increased with the continuous development of vehicle collision avoidance, lane keeping, and autonomous cruise control technologies. Independent sensing units, such as millimeter-wave radar, ultrasonic radar, and vision cameras with satisfactory external sensing capabilities, have gradually been applied to vehicle target detection [1–7]. However, a single type of sensing unit fails to meet the demand for sensing capability in vehicle automation enhancement, and researchers are gradually shifting their focus from single-sensor sensing toward the direction of fusion sensing. The vision camera-based image target detection technology has matured and can be used in practical scenarios due to the collaboration efforts of many researchers [8,9].

Krizhevsky et al. [10] proposed the deep convolutional neural network called AlexNet in the Visual Recognition Challenge, which laid the foundation for research in the field of deep learning target detection. Girshick et al. [11] put forward an R-CNN detection model

based on convolutional neural networks to identify candidate regions where targets may be present. He et al. [12] established the spatial pyramid pooling algorithm SPPNet to solve the fixed-size limitation on input images in R-CNNs. The algorithm outperforms the previously proposed YOLO algorithm [13] in the detection of small targets and improves the detection capability of small targets. Radar imaging has been extensively investigated [14,15]. Chen [16] developed an extended algorithm that can effectively avoid errors caused by energy distribution imbalance. Dong [17] proposed a compressed sensing algorithm for synthetic aperture radar and provided ideas for multi-sensor fusion.

Visual target detection has undergone two stages of development: traditional and deep learning-based target detection algorithms, which remarkably improved the detection accuracy and speed of image targets [18,19]. However, pure vision-based target detection still presents inherent disadvantages in addressing complex scenarios such as multiple target overlap, pedestrian detection in dense traffic, and fog. Millimeter-wave radar shows acceptable detection capability in target detection in terms of estimation of target position, velocity, and other state data. However, using the millimeter-wave radar to perform tasks, such as target class and target lane estimations, presents limitations. Therefore, many researchers have considered the application of multi-sensor fusion methods in target detection.

## 2. Related Work

The goal of multi-sensor fusion target detection is primarily to exploit the complementary nature of multiple sensors, such as detection capability, production, and maintenance costs and stability, under a variety of conditions. Research on fusion between millimeter-wave radar and vision is still in its infancy and is limited by the lack of publicly available datasets containing millimeter-wave radar data [20–24]. Millimeter-wave radar-based fusion algorithms have gradually attracted research attention with the release of the nuScenes dataset and simulation software programs such as CARLA.

Ji et al. [25] created regions of interest for picture target detection with radar detection in a simple neural network for target detection. Many studies have [26–28] also used millimeter-wave radar detection to guide image target detection. Jin et al. [29] achieved detection and recognition of multiple targets on the basis of regions of interest for image detection by exploring the integration of millimeter-wave radar and vision fusion in spatial and temporal dimensions. Song et al. [30] applied millimeter-wave radar with image 3D target detection to divide the sensor task according to the radial distance of the target for multi-sensor-supervised hazardous target detection and classification. Vijay et al. [31] proposed the use of RVNet with millimeter-wave radar structures and camera image data as inputs to the convolutional neural network. Jha et al. [32,33] used independent detection results from millimeter-wave radar and vision sensors for decision fusion algorithms. Lekic et al. [34] utilized a deep learning approach based on adversarial networks to fuse camera and millimeter-wave radar data into a bird's eye view for free-space detection. Chadwick et al. [35] projected millimeter-wave radar data into the image plane, used deep neural networks for detection, and fused radar features with visual features through tandem fusion and achieved satisfactory performance in the authors' self-defined dataset.

Millimeter-wave radar and vision fusion demonstrate advantages in small target detection. Aziz et al. [36] proposed an algorithmic framework for fusing millimeter-wave radar and visual information to achieve target detection. Chang et al. [37] put forward a millimeter-wave radar and vision fusion target detection algorithm based on spatial attention fusion to improve the detection of small and minimally deterministic targets by introducing a spatial attention module. Nabati et al. [38] established a center-based millimeter-wave radar and vision fusion 3D target detection algorithm for millimeter-wave radar and vision target association. Jhon et al. [39] improved the weak detection capability of vision cameras for target detection under night, rain, and fog conditions using millimeter-wave radar, vision cameras, and thermal imaging cameras in a weak detection environment. Nabati et al. [40] obtained a set of detection a priori frames by expanding

from the center of the detection frame to all around the a priori frames for image detection based on the uncertainty of the radar point detection target. Wang et al. [41] used coordinate transformation with edge detection of vehicles to bridge the gap of single-sensor detection of vehicles. Wang et al. [42] fused attentional mechanisms and driver awareness to improve the integrated performance.

Research attention on fusion algorithms based on millimeter-wave radar and vision information is generally focused on the decision layer. This approach is a fusion of independent millimeter-wave radar and vision detection results performed by a set logic. However, the detection rate and environmental applicability still require improvement. Therefore, an extended target detection network with multilayer feature fusion based on millimeter-wave radar and visual raw information is proposed in this study to improve the detection accuracy and robustness of the model under complex weather conditions.

## 3. Method

### 3.1. FPN Extended Network

Researchers have typically predicted features separately from each layer of the detection network, thereby reducing the validity of information in the feature map as the feature depth increases and rendering the algorithm ineffective in small target detection. The proposed feature pyramid network (FPN) can provide a satisfactory solution for target detection from multiscale features. Its main solution is the multiscale problem in object detection. FPN adopts a top–down and lateral connection structure to fuse the underlying location information with a semantic information-rich feature map at high levels. The location information of the underlying target stored in the new feature map is obtained, and the detection of small-scale targets is improved. Depth features obtained from different convolutional layers are matched using $1 \times 1$ convolutional kernels for channel matching, and fused feature maps are processed with $3 \times 3$ convolutional kernels to reduce the effect of confusion caused by the fusion of features from different layers. We use a three-layer FPN as an example and extend it. The FPN structure is shown in Figure 1a.

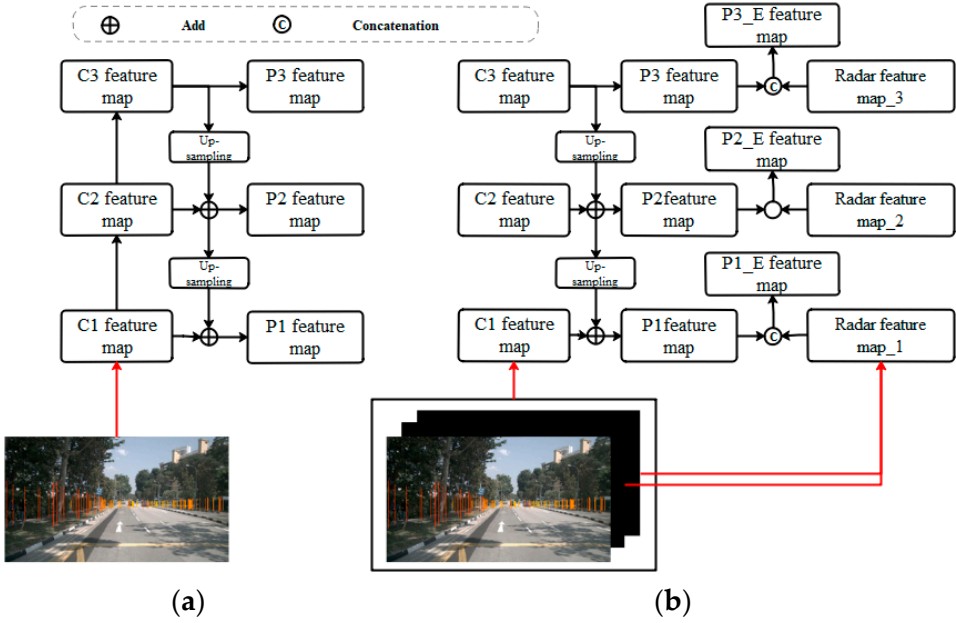

**(a)**            **(b)**

**Figure 1.** Feature Pyramid Network and its extension. C denotes the convolutional layers of a CNN and P denotes prediction layers. (**a**) FPN structure; (**b**) extended FPN (E-FPN).

Deep features are upsampled and fused with shallow features through addition (Add) to obtain semantically enhanced feature maps P1 and P2 for image features C1, C2, and C3 as well as to improve the algorithm's recognition of small targets. The detection results can still be improved on the basis of the FPN and its acceptable performance in

synthesizing multilayer semantic information. We propose the extended FPN (E-FPN) shown in Figure 1b. The proposed algorithm adds millimeter-wave radar features at the corresponding scales for tandem (concatenation) fusion to P1, P2, and P3 feature maps, which are extracted via the FPN to obtain the enhanced P1_E, P2_E, and P3_E fusion feature maps for prediction. We employed a seven-layer FPN in this study to perform sensor data fusion detection tasks. We also extended the VGG network in Section 3.2.

### 3.2. VGG Extended Network Block Design

The VGG network was proposed by the Visual Geometry Group at Oxford as an abbreviation of the laboratory's name. The VGG study illustrates some of its findings from the ImageNet 2014 challenge and suggests that a deep model can be constructed by reusing underlying blocks. Figure 2a shows the block structure of the VGG network. A CNN base block is generally constructed as follows: a convolutional layer; a nonlinear activation function, such as ReLU; and a pooling layer, such as the maximum pooling layer. The method proposed in the VGG study aims to use several consecutive identical convolutional layers with a padding of 1 and a window shape of 3 × 3, followed by a maximum pooling layer with a step size of 2 and a window shape of 2 × 2. The convolutional layer maintains the height and width of the input and output constant, while the pooling layer halves the size of the input. We also propose the extended VGG network apart from the VGG network.

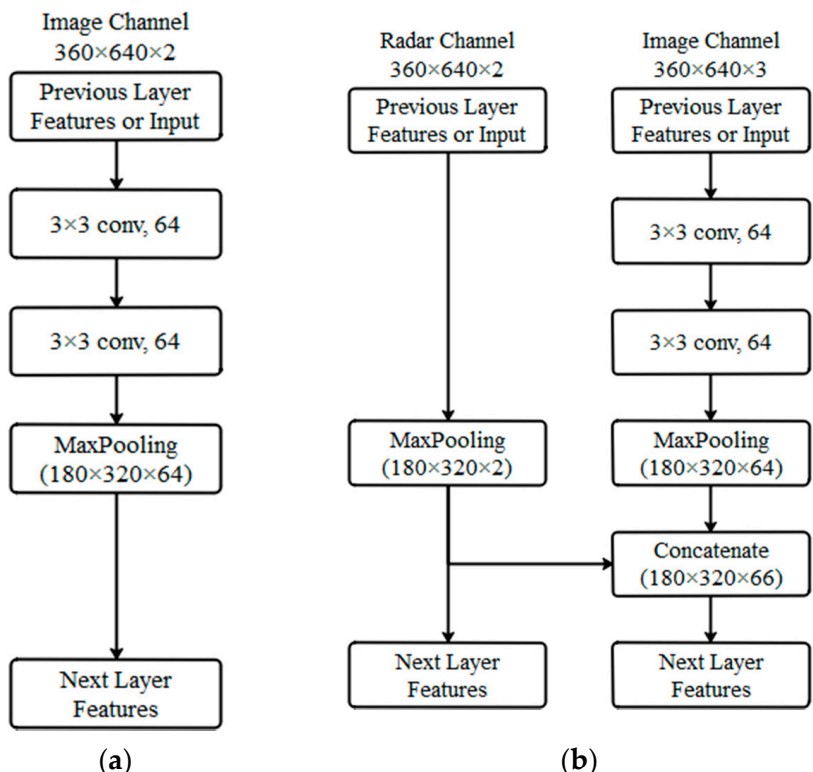

**Figure 2.** VGG network blocks and extension blocks. (**a**) VGG-16 Block. (**b**) Extended VGG Network Block.

Two channels are included in the radar image generated by the extension: radar cross section (RCS) and range channels. The radar image size is the same as the visual image size. The VGG network block structure is redesigned to accommodate the "millimeter-wave radar–visual image" extended image as the input to the network and incorporate radar image features into the convolutional network. Figure 2 presents the block structures of VGG and the extended VGG network. Figure 2b illustrates that the extended VGG network used millimeter-wave radar two-channel images and visual (R, G, B) three-channel images as network inputs. The image channel completes the convolution and pooling of the

original VGG block and concatenates with millimeter-wave radar image features. The obtained feature map is then used as the input of the next-level VGG block.

The advanced architecture of the extended target detection network of millimeter-wave radar and vision fusion based on the RetinaNet with extended VGG-16+E-FPN in this work is shown in Figure 3. The extended VGG-16 backbone detection network structure is presented in the left portion of Figure 3. Meanwhile, the extended FPN structure is illustrated in the dotted box in the middle.

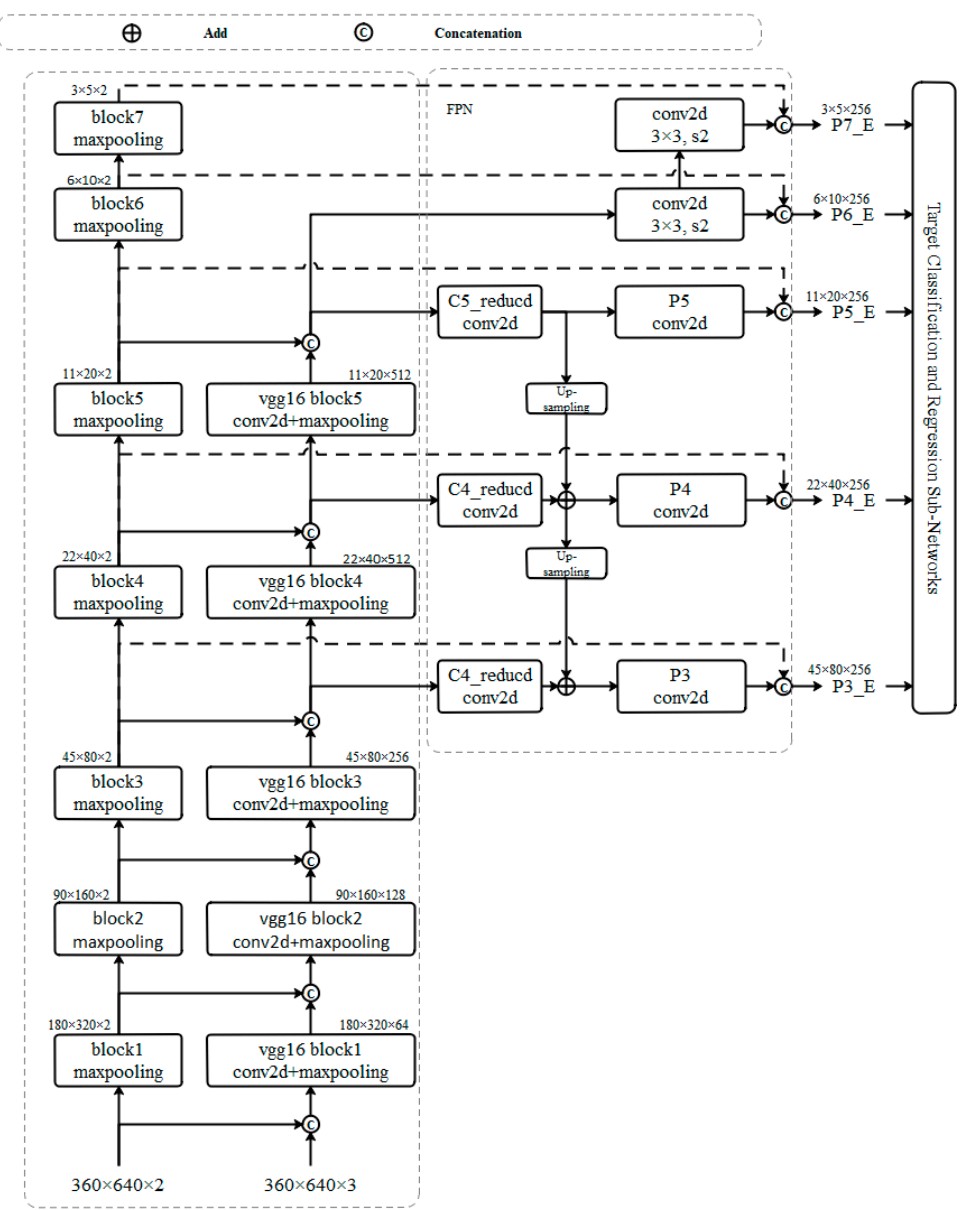

**Figure 3.** Advanced architecture for extended target detection networks in this paper.

Figure 3 shows that the extended network structure regards two channels of millimeter-wave radar images and three channels of visual images as inputs, and cascades visual image features with radar image features as composite features using tandem fusion (number of composite feature channels = number of pretandem radar feature channels + number of pretandem image feature channels). Composite feature outputs from each layer of the extended VGG backbone detection network are C5_reduced, C4_reduced, and C3_reduced and radar feature maps R1, R2, . . . , R7. Fusion and radar features obtained from the VGG backbone detection network are used as inputs in the E-FPN. Meanwhile, extended feature

maps P3_E, P4_E, P5_E, P6_E, and P7_E obtained from the extended pyramid network are utilized as inputs in the target network.

A fundamental difference can be observed in the amount of information contained in a pixel between millimeter-wave radar and image pixel points. Radar images use the target distance as the pixel value, while the target information in visual images must be represented by a single pixel point together with nearby pixel points. The shallow fusion of radar images with visual images is poorly correlated with the information expressed by the two, given that the input information shows minimal semantic similarity and only indirectly associates features. Input data in deep networks can represent increasingly dense semantic information and provide the feature information needed to facilitate the classification task. Therefore, deep convolutional features C5_reduced, C4_reduced, and C3_reduced are selected as feature outputs to ensure the unfolding of multiscale images in the FPN and the semantic similarity between millimeter-wave radar and visual information in deep features. Weights of different radar feature layers can be adjusted accordingly during network training to train the network adaptively and obtain the optimal prediction classification effect by fusing millimeter-wave radar features R1, R2, ... , R7 with pyramid network output features P3_E, P4_E, P5_E, P6_E, and P7_E in the E_FPN in series.

## 4. Result

### 4.1. Evaluation Indicators

The proposed millimeter-wave radar and vision fusion-based target detection algorithm aims to enhance image target detection by guiding the network on the basis of millimeter-wave radar data.

Target classification evaluation in the evaluation of image target detection is based on the average accuracy, while target localization evaluation is based on intersection and merge ratio. The evaluation indicators are presented as follows:

(1)  True Positive (*TP*): Intersection over Union (*IoU*) of prediction and true value boxes is greater than the threshold value, and the classification is correct.

*IoU* is the result obtained by dividing the overlapping part of two regions by the aggregated part of the two regions. The result is compared with the findings of the *IoU* calculation by means of a set threshold value. The *IoU* is defined as follows:

$$IoU = \frac{Area\ of\ Overlap}{Area\ of\ Union} \tag{1}$$

(2)  False Positive (*FP*): The prediction box contains the target but a truth box is absent or the *IoU* with the truth box is less than the threshold.
(3)  False Negative (*FN*): A missed detection situation where no prediction is made for the real target although the target is real.
(4)  Precision: The proportion of targets correctly classified among all detected targets with *IoU*s meeting the threshold.
(5)  Recall: The proportion of detected targets with correct classification, and the *IoU* is greater than the threshold among all truth targets.

The accuracy and recall rates are calculated as follows:

$$Precision = TP\ /\ (TP + FP)$$
$$Recall = TP\ /\ (TP + FN) \tag{2}$$

### 4.2. Dataset Introduction

Autonomous driving datasets have been typically designed around visual images and LIDAR raw data. The lack of millimeter-wave radar data in public datasets and the low availability of self-built datasets have hindered the development of convergence, to some extent. This problem was gradually alleviated when the nuScenes dataset [43] was released in 2019. The nuScenes dataset is a large-scale public dataset for autonomous driving

developed by the Motional team. The dataset was collected in Boston and Singapore with 1000 driving scenes of about 20 s each.

Sensor data acquisition is also performed in this work through the HYPERVIEW smart driving car platform for testing and validation of the algorithm. Figure 4 shows the physical diagram of the whole vehicle acquisition platform containing millimeter-wave radar, vision camera, and LIDAR.

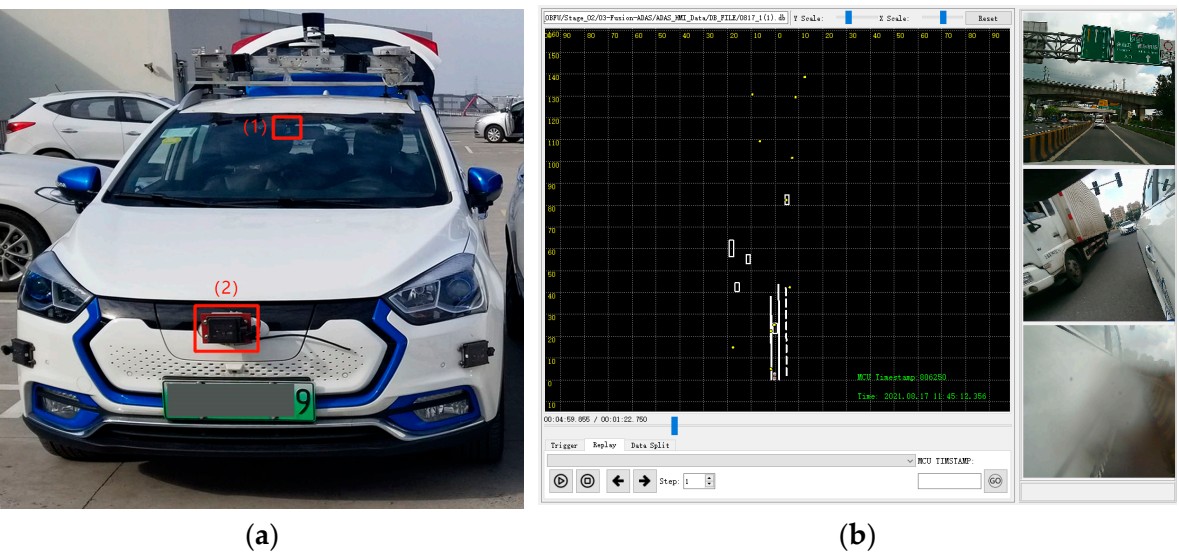

(**a**)  (**b**)

**Figure 4.** nuScenes data acquisition platform sensor configuration. (**a**) Vehicle collection platform. (1) is a camera (2) is a millimeter wave radar. (**b**) HMI tool interface.

ProtoBuf and SQLite data compression is used for the self-built data acquisition platform for millimeter-wave radar and visual image acquisition during the road test. Specific parameters of the millimeter-wave radar are listed in Table 1.

**Table 1.** Continental ARS410 mm-wave radar operating parameters.

| Range | Sampling Frequency | Working Frequency | Speed Accuracy |
|---|---|---|---|
| <170 m | 20 Hz | 76–77 GHz | ±0.1 km/h |

The data acquisition and playback HMI tool interface are shown in Figure 4b. This tool is used to acquire millimeter-wave radar-structured data and visual images for real-time display while storing self-vehicle motion information and sensor timestamps.

### 4.3. Comparison of the Detection Effect of Converged Network and Reference Network

Images in the nuScenes dataset and the generated radar image width and length are adjusted to 360 × 640 as the input of the network. The subjective detection results in the nuScenes and self-constructed datasets are presented in this section.

Obstacles detected by the millimeter-wave radar in the technical application of in-vehicle millimeter-wave radar are distributed in the top-view plane within the millimeter-wave radar field of view (FOV). Hence, it does not contain the physical coordinates of the target in the vertical direction and it can be projected into the 2D plane for visualization and data representation. The classical coordinate transformation method in the small-aperture effectiveness model is chosen for the projection method. It can be represented in the image and pixel planes in the form of point cloud images for mapping.

Figure 5 shows the comparison of the target detection effect of the fusion network with the reference network performed on the test set of the nuScenes dataset. Figure 6 presents the comparison of the target detection effect of the fusion network with the reference network performed on the homemade dataset. The reference network used

in Figures 5 and 6 is the RetinaNet target detection network based on the VGG-16+FPN backbone detection network. Each column in Figure 5 corresponds to a scene. The first row shows the projection of the millimeter-wave radar point cloud used for prediction on the visual image. The second row indicates the feature output of the first extended VGG detection block of the extended target detection network and the feature output of the first VGG detection block of the reference network. The third row presents the feature difference between the first detection block of the extended network and the reference network. The fourth row represents the feature output of the extended target detection network P3_E and the reference network P3. The fifth row depicts the feature difference between the extended network and the reference network FPN-P3 (P3_E). The sixth row shows the feature output of the extended target detection network P5_E and the reference network P5.

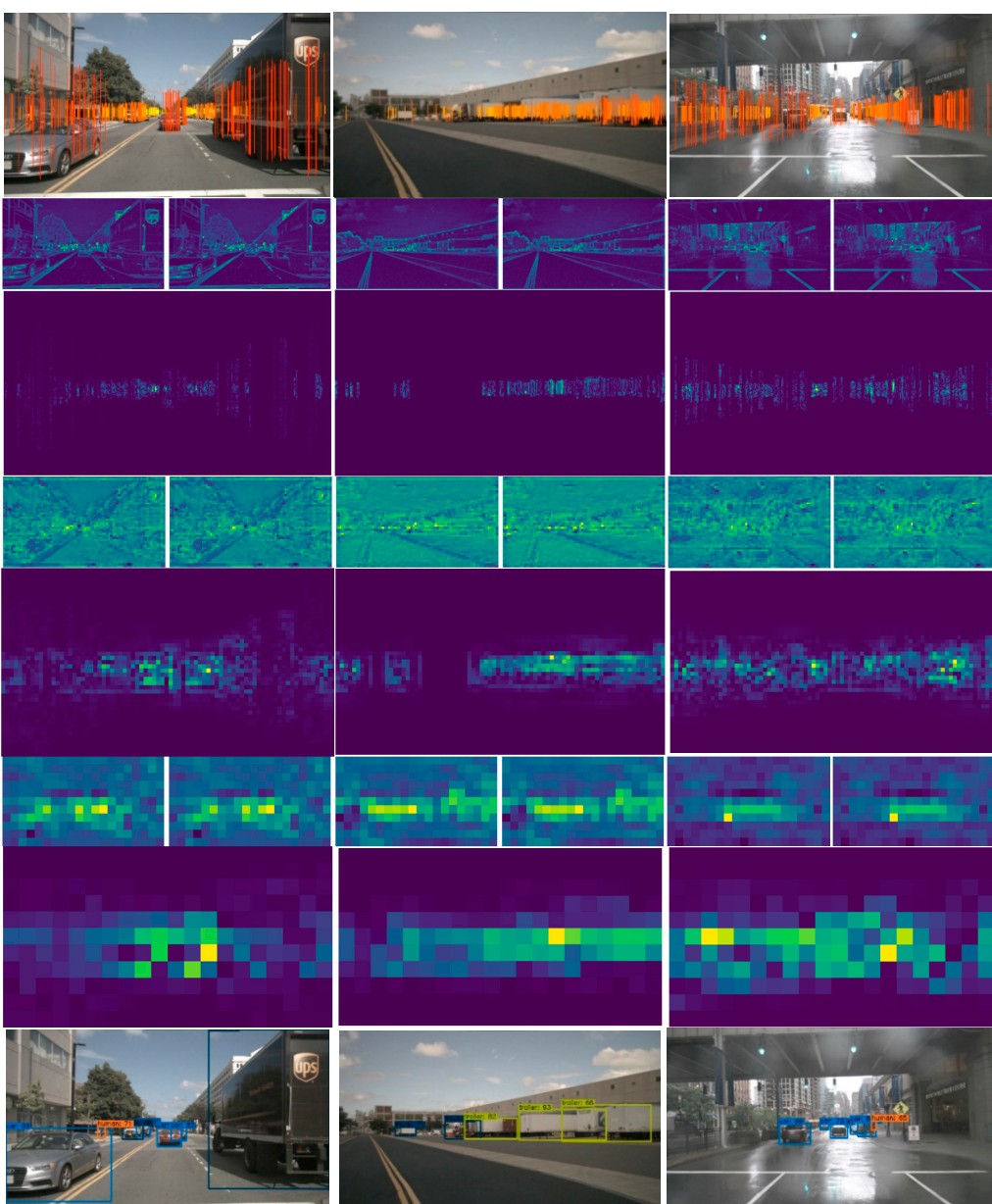

**Figure 5.** *Cont.*

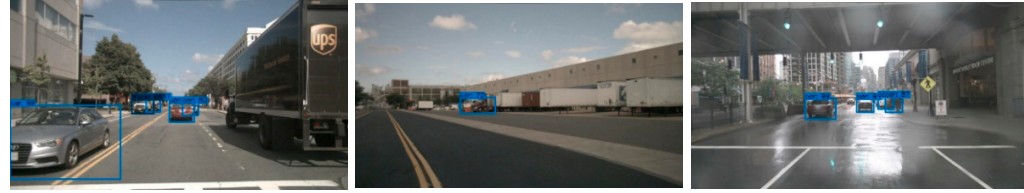

**Figure 5.** Comparison of Detection Results on the NuScenes Dataset.

**Figure 6.** Comparison of Detection Effects on Data Collection Platforms.

The seventh row presents the feature difference between the extended network and the reference network FPN-P5 (P5_E). The eighth row exhibits the detection effect of the proposed extended fusion target detection network. Finally, the ninth row demonstrates the detection effect of the reference network.

According to the comparative analysis in Figure 5, the fusion target detection network shows an enhanced recognition and classification effect on difficult-to-identify feature scenarios, such as small targets at long distances (e.g., pedestrians in shadows), multiple targets with similar overlapping textures (e.g., multiple white vans shown in the second column), and rain-obscured pedestrian targets (with features superimposed at different distances in the radial direction according to the radar extended image). The small pedestrian target detected by the first column of the fusion network is used as an example. The location of the pedestrian presents evident radar output features at the position corresponding to the feature difference images of P3_E and P3. Meanwhile, the large target van detected by the second column of the fusion network is used as an example. Multiple white vans overlap and demonstrate inconspicuous texture contours, which cannot be recognized and classified in the image target detection through the reference network but can be applied to obtain satisfactory detection results in the fusion network on the basis of radar image channels.

The fusion target detection network outperforms the reference network in the recognition of small long-range targets in night scenes and overcast environments (Figure 6) and the detection of small long-range targets in the homemade dataset. The feature enhancement of the P3 (P3_E) feature at the location of the detected small target in Figure 6 demonstrated that the addition of the millimeter-wave radar image channel information can help the detection process by strengthening long-range small target and independent feature recognition of overlapping vehicles shown in the second column.

*4.4. Comparison of the Effect of Continuous Detection in a Variety of Scenes*

4.4.1. Daytime Complex Scene Target Detection

Target detection effects of three consecutive frames of the daytime scene are compared in Figure 7. The first row shows the effect of the proposed extended fusion target detection. The second row presents the effect of the target detection of the reference network. The figure demonstrates that the extended fusion target detection network in this work enhances the detection continuity for small distant targets. A satisfactory detection effect can be obtained for small fuzzy targets (vehicle) on the left side on the basis of radar point cloud feature enhancement, and detection and classification can be achieved for large trucks in close view with incomplete image information.

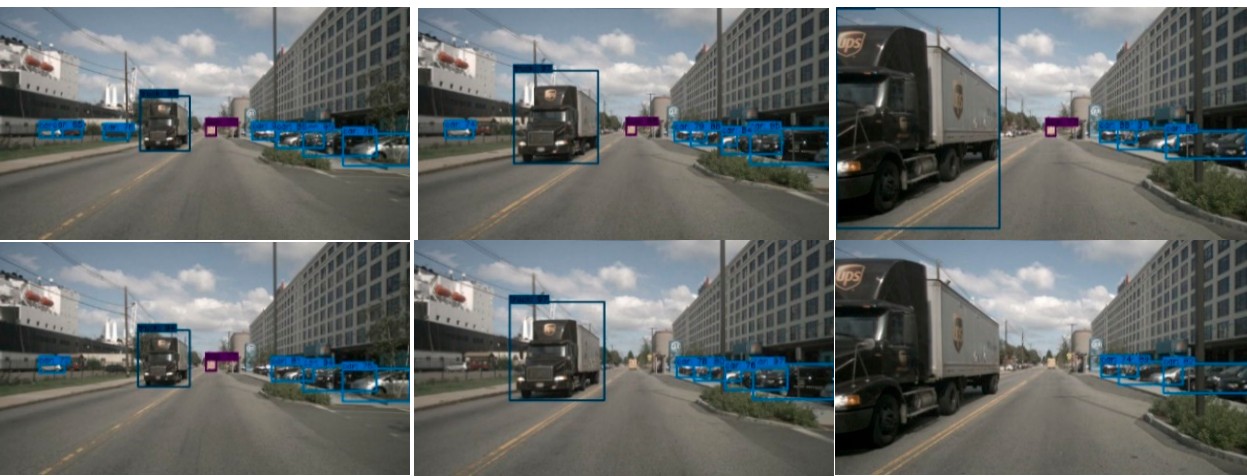

**Figure 7.** Comparison of Continuous Detection Effects of Daytime Scenes in the nuScenes Dataset.

As shown in Figures 8 and 9, a vehicle gradually approaches the self-vehicle to the meeting process in the night scene. The first row shows the target detection effect of the extended fusion network and the second row demonstrates the target detection effect of the reference network in each set of images. Target detection is achieved three detections earlier in the process of target approach through the extended fusion perception network compared with that using the reference network (Figure 8). The reference network produces classification errors during target detection and loses the ability to detect the target because it gradually drives away from the field of view during the meeting process shown in Figure 9.

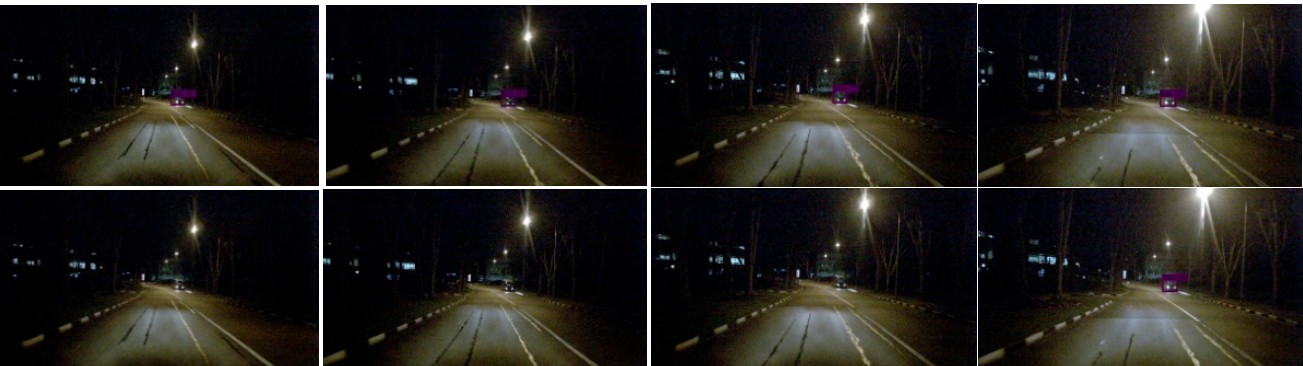

**Figure 8.** Comparison of Nighttime Scenes in the nuScenes Dataset. Continuous Detection of Target Presence.

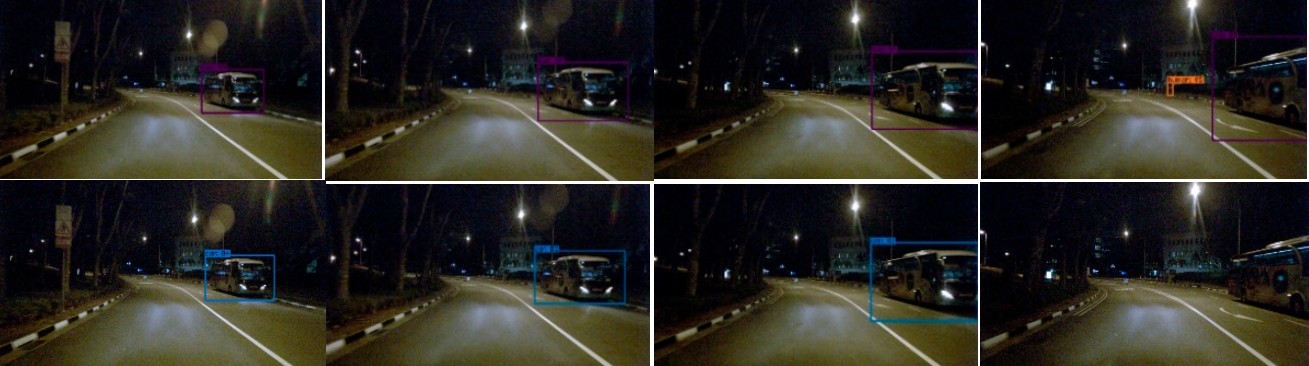

**Figure 9.** Comparison of Nocturnal Scene. Target Departure Sequential Detection Effects on nuScenes Dataset.

4.4.2. Nighttime Complex Scene Target Detection

The image in the first row of Figure 9 shows that the fused sensory detection network can obtain the detection of distant pedestrian targets. It can improve the detection capability of small targets in harsh environments and reduce the influence of possible accident hazard targets on the motion state of self-vehicles.

*4.5. Converged and Reference Network Detection Capability Analysis*

The proposed extended fusion and reference networks for different classes of targets were tested and statistically analyzed by randomly selecting 25 scenes from a total number of 150 scenes in the v1.0-test subset of the nuScenes dataset. The analysis results are shown in Figure 10.

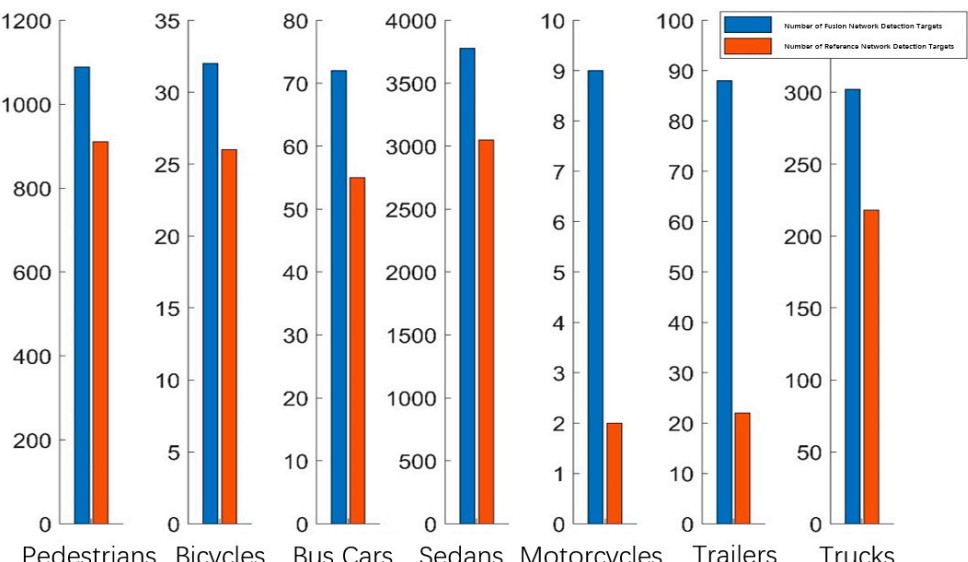

**Figure 10.** Fusion Network and Reference Network Detection Target Statistics.

The total number of detection acquisition targets was determined for each category (Figure 10).

The extended fusion network presents an overall improvement of 25.36% in the number of detected targets compared with the reference network, and the improvement in the detection capability of large sample targets, such as cars and pedestrians, is around 20% (Table 2). The comparative analysis of target detection effects described in the previous section showed that the fusion target detection algorithm based on the proposed extended network significantly improves the detection effect compared with that of a single type of sensor.

**Table 2.** Structure of the millimeter-wave radar data from the nuScenes dataset.

| Category | Pedestrians | Bicycles | Bus Cars | Sedans | Motorcycles | Trailers | Trucks | Total |
|---|---|---|---|---|---|---|---|---|
| Number of converged network detections | 1089 | 32 | 72 | 3777 | 9 | 88 | 302 | 5369 |
| Number of reference network detections | 911 | 23 | 55 | 3049 | 2 | 22 | 218 | 4283 |
| Effectiveness enhancement Percentage (%) | 19.53 | 23.07 | 30.91 | 23.88 | – | 300 | 38.53 | 25.36 |

*4.6. Objective Analysis of Values*

We verified the practical effectiveness and detection accuracy of the proposed extended fusion target detection algorithm by extracting 20% of the algorithm from the v1.0-trainval subset of the nuScenes dataset and using them as the validation set.

Figure 11 illustrates the statistical analysis of the average accuracy of the proposed extended fusion target detection (solid part) and reference (dashed part) networks in different categories. The overall average accuracy (mAP) of the two algorithms is calculated, compared, and then plotted with curves.

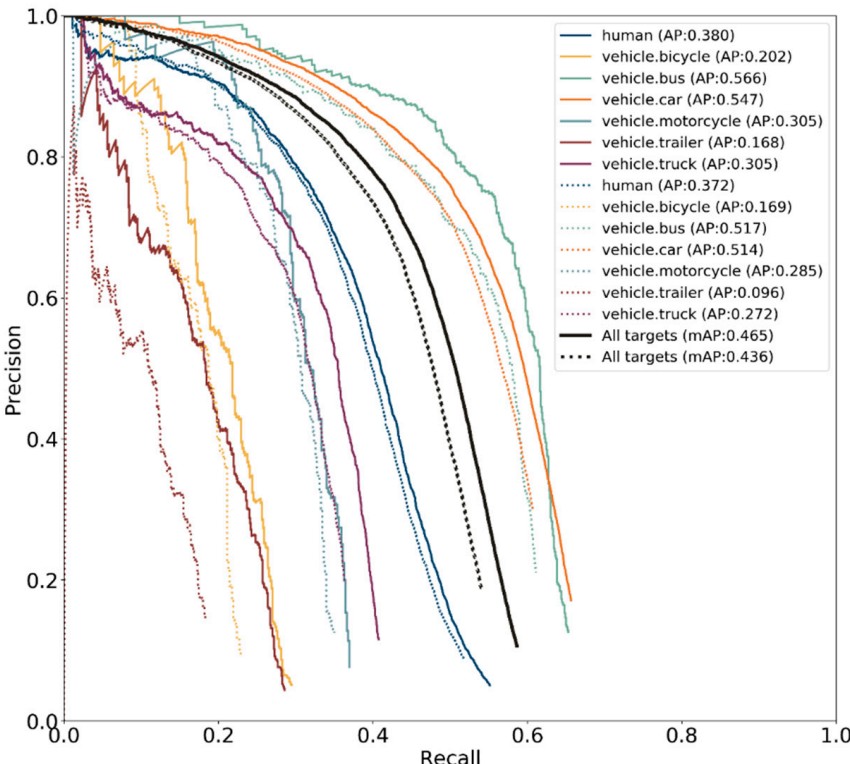

**Figure 11.** Comparison of Extended and Reference Networks Tested by Category in the nuScenes Dataset. Solid lines represent the proposed technique, and dotted lines represent the reference network.

The comparison of the average accuracy of different algorithms by category on the nuScenes dataset is presented in Table 3. RRPN is a millimeter-wave radar and vision fusion target detection network. Fast R-CNN and baseline are pure vision target detection networks. The comparison is illustrated with these algorithms because the nuScenes dataset contains only a few 2D target detection algorithms. The authors use real-time millimeter-wave radar data in the RRPN algorithm to generate region suggestion frames instead of the selective search algorithm in the fast R-CNN algorithm to generate detection frames. This approach improves the accuracy and recall and remarkably reduces the time consumption of the selective search algorithm.

**Table 3.** Comparison of the different algorithms by category AP on the nuScenes dataset.

|            | mAP       | Pedestrians | Bicycles  | Bus Cars  | Sedans    | Motorcycles | Trailers  | Trucks    |
|------------|-----------|-------------|-----------|-----------|-----------|-------------|-----------|-----------|
| RRPN       | 0.430     | 0.220       | **0.306** | 0.664     | 0.442     | **0.434**   | -         | 0.516     |
| Fast R-CNN | 0.418     | 0.155       | 0.241     | **0.722** | 0.472     | 0.354       | -         | **0.545** |
| Baseline   | 0.436     | 0.372       | 0.169     | 0.517     | 0.514     | 0.285       | 0.096     | 0.272     |
| Paper      | **0.465** | **0.380**   | 0.202     | 0.566     | **0.547** | 0.305       | **0.168** | 0.305     |

The analysis of mAP metrics demonstrated that the proposed extended network-based fusion target detection algorithm in this work improves mAP by 3.5%, 4.7%, and 2.9% compared with the RRPN, Fast R-CNN image target detection network, and reference network in this work, respectively (Table 3). According to the analysis of AP metrics by category, the proposed detection algorithm achieves the best performance in the nuScenes dataset among several common targets on the road, such as pedestrian, car, and trailer targets. The proposed feature-level fusion target detection network based on extended fusion that generates detection frames on the basis of regions of interest significantly outperforms the RRPN algorithm from the perspective of fusion algorithms. This work

uses the following methods according to the definition of AP at different scales in the COCO dataset to verify the improvement effect of the proposed fusion target detection method on the detection ability of targets at various scales.

(1)  Aps: Average accuracy of small-scale targets with prediction frame area less than $32^2$.
(2)  APm: Average accuracy of medium-scale targets with prediction frame area within $[32^2, 96^2]$.
(3)  APl: Average accuracy of large-scale targets with prediction frame area greater than $96^2$.

This work was tested against the v1.0-trainval training validation set of the nuScenes dataset. Figure 12 and Table 4 show that the proposed algorithm outperforms the target detection algorithm of the vision-based reference network on all scales. The average accuracy for small targets is improved by 18.73%, which is significantly higher than the detection capability for medium and large targets.

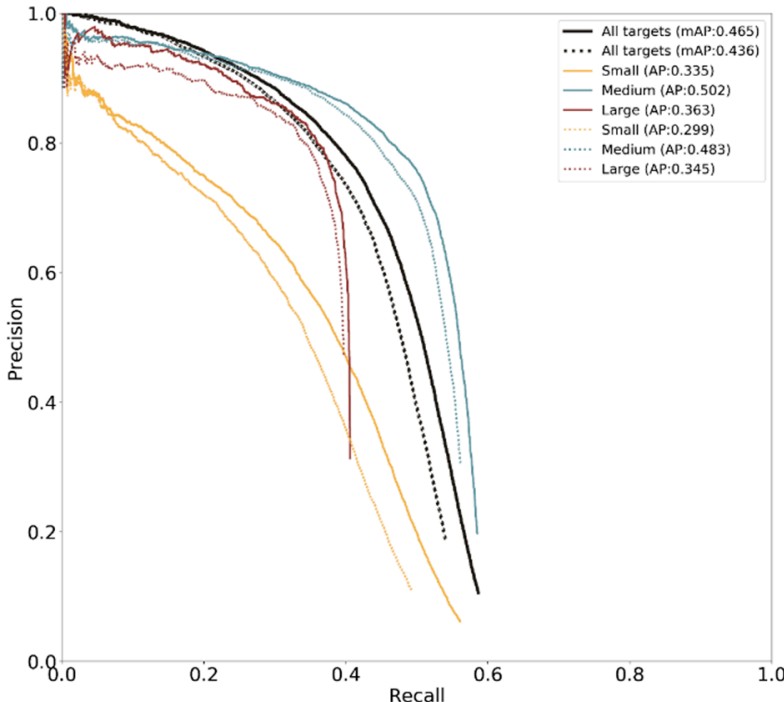

**Figure 12.** Comparison of the extended network of nuScenes dataset with the reference network tested by scale. Solid lines represent the proposed technique, and dotted lines represent the reference network.

**Table 4.** Comparison of the extended converged network in this paper with the reference network by scale AP.

|  | mAP | APs | APm | APl |
|---|---|---|---|---|
| Baseline | 0.436 | 0.299 | 0.483 | 0.345 |
| Paper | **0.465** | **0.355** | **0.502** | **0.363** |
| Performance boost | **6.65%** | **18.73%** | **3.93%** | **5.22%** |

## 5. Discussion

The driving environment in which smart cars are located is different from the working environment in which other artificial intelligence machines are located, with high speed and complexity. Cameras used to acquire image data are susceptible to light, and LIDAR, which acquires point cloud data, is susceptible to harsh environments. Millimeter-wave radar is not good at detecting stationary targets, and the defects of the sensors themselves make it

impossible for smart cars to perform sensing tasks with only a single sensor. Because LIDAR is expensive, this paper focuses on how to achieve surrounding environment detection by fusing millimeter-wave radar data with camera data. A reliable sensing system is a prerequisite for a smart car to operate properly under complex traffic conditions, and the researchers hope that the multi-sensing technology can not only improve the detection accuracy, but can also be robust. For example, the sensing system can still operate properly in a low-light environment where the detected objects are too small. Thus, this paper proposes a target detection algorithm that fuses millimeter-wave radar with camera data. First, based on the analysis of feasible technical routes for millimeter-wave radar and vision fusion, the fusion route in this paper is determined as feature-level fusion, and the training and test evaluation of the algorithm are carried out based on the nuScenes dataset. Then, millimeter-wave radar images are introduced as auxiliary information for visual image target detection. In this paper, the scalability of the RetinaNet one-stage target detection algorithm based on the VGG-16+FPN backbone detection network is investigated. The extended VGG-16 network and extended feature pyramid network (E-FPN) applicable to millimeter-wave radar and vision fusion are proposed through the study of VGG-16 and feature pyramid networks. The deep extended fusion target detection network for millimeter-wave radar and vision fusion is proposed by performing tandem fusion of millimeter-wave radar features and visual features in each layer of the backbone detection network. By training and testing the proposed network on the nuScenes dataset, compared with the (VGG-16+FPN) RetinaNet reference network (Baseline) for pure visual image target detection, the mAP of the proposed network in this paper is improved by 2.9% and the small target detection accuracy is improved by 18.73%, which verifies the extended fusion target detection network proposed in this paper. The detection capability and algorithmic feasibility of the extended fusion target detection network proposed in this paper for visually insensitive targets are also verified. The research in this paper has a certain reference value for the research of multi-sensor fusion perception technology for smart driving vehicles, especially in the field of fusion target detection, which is of positive significance for the research of breaking the bottleneck of traditional single-sensor detection capability and improving the comprehensive perception performance of complementary sensors.

## 6. Conclusions

The current single-sensor-based perception and logic-based heterogeneous sensor fusion can no longer meet the in-depth needs of high-level autonomous driving for environmental sample information. The market-oriented application of smart driving cannot hope for a substantial improvement of single-sensor detection capability in the short term with the development of sensor technology research and target detection technology entering a bottleneck period. Based on smart driving cars, with millimeter-wave radar and vision detection as research objects, this work proposes an extended fusion target detection network by conducting research on multi-sensor fusion target detection algorithms to cope with the higher demand for intelligent sensing capabilities in urban environments and high-speed road environments where high-level smart driving is taking place. In this work, the RetinaNet algorithm based on the VGG-16+FPN backbone detection network is extended to address the shortcomings of the RetinaNet one-stage target detection algorithm in terms of detection accuracy. The extended VGG-16 network and E-FPN suitable for multi-channel input of millimeter-wave radar images and visual images are proposed. The millimeter-wave radar feature extraction network and visual target detection network are introduced for deep fusion, and the network is trained in the nuScenes dataset. Based on the validation of the proposed network and the comparison with the reference network, the proposed network improves the target detection rate by about 25%, the mAP by 2.9%, and the average accuracy of small targets by 18.73%. The experiments in this work validate the feasibility of the proposed extended fusion target detection network based on millimeter-wave radar filtering algorithms in enhancing the capability and technical

approach in traffic environment target detection applications, especially for small targets, which is greatly improved.

**Author Contributions:** Conceptualization, S.S. and X.W.; methodology, C.Q., N.Z. and C.S.; software, C.Q., S.S. and F.X.; validation, C.Q., F.X. and S.S.; formal analysis, C.S. and X.W.; investigation, N.Z., S.S. and C.Q.; resources, F.X.; data curation, C.Q., N.Z., provide suggestions for the revision. All authors have read and agreed to the published version of the manuscript.

**Funding:** This work was supported in part by the National Key R&D Program of China under Grant 2021yfb2500704, Science and Technology Development Plan Program of Jilin Province under Grant 20200401112GX, and in part by the Industry Independent Innovation Ability Special Fund Project of Jilin Province under Grant 2020C021-3.

**Institutional Review Board Statement:** Not applicable.

**Informed Consent Statement:** Not applicable.

**Data Availability Statement:** Not applicable.

**Acknowledgments:** The authors would like to express their gratitude to the editors and the anonymous reviewers for their insightful and constructive comments and suggestions, which have led to this improved version of the paper.

**Conflicts of Interest:** The authors declare no conflict of interest.

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
