# Peer review of "Millimeter-Wave Radar and Vision Fusion Target Detection Algorithm Based on an Extended Network"

_machines, doi:10.3390/machines10080675_

Round 1

Reviewer 1 Report

1. It appears the authors started with a suggested paper content format and forgot to remove the suggestions (need to remove statements not related to this paper. Example: remove lines 36-44 in section 1, lines 122-124 in section 3, line 143 in section 3.2, line 201-204 in section 4 and so on)

2. The introduction of FPN in section 3.1 needs to be rewritten. It is written in way that the authors assumes the reader is aware of every details of FPN. The authors simply introduces feature maps P1, P2 and image features C1, C2 etc., without mentioning what they are in the pyramid. A better way would be to introduce C's as the convolutional layers of a CNN and then P's as prediction layers. Further, the FPN was introduced in this section having only 3 layers as if that is the definition of FPN. Rather authors should mention that this is an example of FPN (the authors themselves use 7 layers later in the paper)

3. Similarly, VGG is poorly introduced. The acronym VGG is never defined what it stands for. There is no description in the body what is figure 2(a). It just appeared to be an oddball in the figure. With respect to Fig. 2, what is Block1. The reader has to go down to Fig. 3 to interpret what the authors meant in Fig. 2.

4. Fig. 2 shows VGG with 5 layers (blocks) whereas Fig. 3 has 7 blocks. Definitely, the authors did not use the VGG described in Fig. 2. They should clarify that they are using Fig. 2 to describe the concept of VGG not to introduce the network that is used in this paper.

5. In lines 144-145, the authors mentioned "two channels of millimeter wave radar". They need to define what these two channels are. Unlike R,G,B channels in camera/video, there is no standard definition of channels in radar output. Radar output is three-dimensional (range, Doppler, angle). Further the output could be complex signal as well. So the authors need to clearly define what they mean by the each of the two channels (which dimensions these two channel represent; and they represent the signal magnitude or RCS or SNR or something the radar provides them or just a 1/0 indicating detection)

6. Fig. 2(b) on the image channels it shows 360x640x5: 5 channels of image? Is this a typo.

7. lines 186-188: "Accordingly, millimeter wave radar images are better suited to meet the information needs of pixel level autonomous driving than visual images." Not sure what is meant by this statement. Camera gives the best pixel resolution in cross-range (angle) but does not provide down-range (distance) information. Radar provides the latter better but is poor in cross-range. Is this what the authors mean? This line of reasoning need to be better explained in the paper

8. line 212: IOU is first introduced: no definition provided

9. lines 305-307: How the projection is done need to be described. There are various techniques for such projections.

10. Figs 11-12: It would be better to indicate on the figures themselves that the dotted lines are for reference and the solid line are for the proposed technique.

11. General comment: The texts in the figures are not easy to read. Better fonts are needed there.

Author Response

Thank you very much for the reviewer's in-depth evaluation, which is very helpful for the improvement of this paper. Our reply is already in the attachment

Reviewer 2 Report

Please see the attached PDF file.

Author Response

(The authors gave the same response as above.)

Round 2

Reviewer 2 Report

Please see the attached pdf file.
